# Development and Validation of a Rapid LC-MS/MS Method for Quantifying Alvocidib: In Silico and In Vitro Metabolic Stability Estimation in Human Liver Microsomes

**DOI:** 10.3390/molecules28052368

**Published:** 2023-03-04

**Authors:** Mohamed W. Attwa, Haitham AlRabiah, Adnan A. Kadi

**Affiliations:** Department of Pharmaceutical Chemistry, College of Pharmacy, King Saud University, Riyadh 11451, Saudi Arabia

**Keywords:** alvocidib, intrinsic clearance, metabolic stability, acute myeloid leukemia, LC-MS/MS, in vitro half-life

## Abstract

Alvocidib (AVC; flavopiridol) is a potent cyclin-dependent kinase inhibitor used in patients with acute myeloid leukemia (AML). The FDA has approved orphan drug designation to AVC for treating patients with AML. In the current work, the in silico calculation of AVC metabolic lability was done using the P450 metabolism module of the StarDrop software package, that is expressed as a composite site lability (CSL). This was followed by establishing an LC-MS/MS analytical method for AVC estimation in human liver microsomes (HLMs) to assess metabolic stability. AVC and glasdegib (GSB), used as internal standards (IS), were separated utilizing a C18 column (reversed chromatography) with an isocratic mobile phase. The lower limit of quantification (LLOQ) was 5.0 ng/mL, revealing the sensitivity of the established LC-MS/MS analytical method that exhibited a linearity in the range 5–500 ng/mL in the HLMs matrix with correlation coefficient (R^2^ = 0.9995). The interday and intraday accuracy and precision of the established LC-MS/MS analytical method were −1.4% to 6.7% and −0.8% to 6.4%, respectively, confirming the reproducibility of the LC-MS/MS analytical method. The calculated metabolic stability parameters were intrinsic clearance (CL_int_) and in vitro half-life (t_1/2_) of AVC at 26.9 µL/min/mg and 25.8 min, respectively. The in silico results from the P450 metabolism model matched the results generated from in vitro metabolic incubations; therefore, the in silico software can be used to predict the metabolic stability of the drugs, saving time and resources. AVC exhibits a moderate extraction ratio, indicating reasonable in vivo bioavailability. The established chromatographic methodology was the first LC-MS/MS method designed for AVC estimation in HLMs matrix that was applied for AVC metabolic stability estimation.

## 1. Introduction

Acute myeloid leukemia (AML) is the most frequent acute leukemia in adults (80%) and it is often difficult to be treated, especially in the elderly (65 years or older) [1,2]. The standard treatment procedure for AML is split into two phases: induction therapy followed by consolidation therapy [3]. Induction therapy composed of a “seven + three” regimen that includes seven days of Cytarabine continuous infusion then three days of the daunorubicin and anthracycline [4]. Unfortunately, AML patients still experience therapeutic side effects and relapse after the complete remission [1,5]. Luckily, numerous novel therapeutics (e.g., Vyxeos, Venetoclax, and Midostaurin) for the treatment of AML have been approved since 2017 [5,6]. However, treatment outcomes remain bleak for older AML patients (65 years or older), with a 5 years’ survival rate [7].

Alvocidib (AVC; flavopiridol, see Figure 1) is small synthetic molecule that is used as a potent cyclin-dependent kinase inhibitor for patients suffering from intermediate- or high-risk AML, who have a poor prognosis [8]. AVC have shown promising therapeutic potential against AML. The FDA has approved orphan drug designation to AVC for the treatment of patients with AML. AVC has been tested in clinical trials under both single and combination scenarios; numerous single-agent Phase II and Phase I clinical trials against solid tumors, leukemia, and lymphomas are currently active [9,10,11]. Unfortunately, almost 50% of the patients in AVC clinical trials exhibited severe adverse effects [12].

A literature review revealed several analytical methodologies for AVC quantification, either alone or with other drugs, including liquid chromatography tandem mass spectrometric (LC-MS/MS) methods [13,14], reversed-phase liquid chromatography electrochemical detection [15] and one high performance liquid chromatography with ultraviolet detector (HPLC-UV) method [16], that were reported for the estimation of AVC. The current LC-MS/MS method exhibited reasonable precision and accuracy (˂6.7%) compared to the reported analytical methods (˂13%) [14,16]. The mean extraction recovery of AVC in the established LC-MS/MS method was 101.4 ± 2.6%, which is better than the reported analytical method (88.6–90%) [14,15]. In addition, the published method utilized a selective reaction monitoring with one mass transition (SRM: 402→341) for detection and a gradient mobile phase system for analysis [14], which is less selective than the multiple reaction monitoring (MRM: 402→341 and 402→70) and isocratic elution mobile phase system that are utilized in the current LC-MS/MS method. No LC-MS/MS analytical method has been reported for the metabolic stability quantification of AVC in human liver microsomes (HLMs).

In the established LC-MS/MS method, isocratic mobile phase was used. A run time and flow rate was 3 min and 0.3 mL/min, respectively, were used, so the methods were rapid and eco-friendly. Moreover, the constructed calibration curve exhibited a linearity in the range of 5 to 500 ng/mL. AVC was tested for its metabolism lability in the HLMs using StarDrop’s WhichP450 model before the starting of the practical experiment to confirm the importance of developing an LC-MS/MS method and to save resources and time. The established LC-MS/MS method was applied for the evaluation of the intrinsic clearance (Cl_int_) and in vitro half-life (t_1/2_) of AVC. In silico StarDrop’s WhichP450 model software and in vitro LC-MS/MS experiments were used for the evaluation of AVC metabolic stability so as to provide more information about the metabolic rate of AVC and to allow in vivo bioavailability assessment.

The current work was performed to develop a rapid LC-MS/MS analytical method for AVC quantification in a HLMs matrix. Our group published other analytical LC-MS/MS chromatographic methods that were applied previously in the computation of the intrinsic clearance (CL_int_) and the in vitro half-life (t_1/2_) of other tyrosine kinase inhibitors [17]. The metabolic stability of a drug is its vulnerability to metabolism and is stated as in vitro t_1/2_ and CL_int_, which are defined as the liver’s ability to metabolize the drug in the blood and the time needed for the 50% metabolism of the parent drug, respectively [18,19]. The in vitro Cl_int_ and t_1/2_ in HLMs were computed by an “in vitro t_1/2_” approach using the “well-stirred” model, as it is the most frequently used model in drug metabolism experiments owing to its simplicity [20,21]. These parameters (Cl_int_ and in vitro t_1/2_) could be used to compute various physiological parameters (e.g., in vivo t_1/2_ and liver clearance). The drug bioavailability evaluation provides a good approach for understanding its in vivo metabolic reactions. If the examined drug shows a fast metabolic rate, it will exhibit low in vivo bioavailability value and a short window of action [22,23,24,25].

## 2. Results and Discussion

### 2.1. In Silico AVC Metabolic Stability

The P450 metabolism model of StarDrop’s software (Optibrium Ltd.; Cambridge, MA, USA) predicts the regioselectivity of metabolism by seven of the major drug metabolizing isoforms of Cytochrome P450 enzymes (CYP3A4, CYP2D5, CYP2C9, CYP1A2, CYP2C19, CYP2C8 and CYP2E1) [26,27,28]. The results are expressed as composite site lability (CSL) value, revealing the AVC metabolic lability. The WhichP450 model predicts the major metabolizing isoform (CYP3A4) for the AVC metabolism, as approved by the pie chart (Figure 2A). The regioselectivity map indicates the predicted sites of metabolism for AVC (Figure 2B). The metabolic landscape (Figure 2C) predicts the AVC metabolic lability of the active sites to improve the understanding of the AVC metabolic rate. These were classified from the highest degree of metabolic instability (labile; orange) to the lowest degree of metabolic stability (mod. labile; yellow color) or metabolic stable (stable; black color). The systematic name of AVC is 2-(2-Chlorophenyl)-5,7-dihydroxy-8-[(3S,4R)-3-hydroxy-1-methyl-4-piperidinyl]-4H-chromen-4-one. From the metabolic landscape of AVC (Figure 2), the C1 of the methyl piperidine group is labile in the metabolism, whereas positions C3 and C7 of the methyl piperidine group and C22, C23, and C24 of the chlorophenyl ring are moderately labile in the metabolism. These outcomes and CSL (0.9975) reflect elevated AVC metabolism instability; so, the established analytical LC-MS/MS method was applied for AVC metabolic stability assessment (Figure 2). The proposed metabolic instability of AVC may be due to the methyl piperidine group, as suggested by the P450 metabolism model of the StarDrop software. These results indicate the value of performing an in vitro metabolic stability experiment for AVC and the need for an LC-MS/MS analytical method for the quantification of AVC in HLMs matrices.

### 2.2. LC-MS/MS Method Development

GSB was selected as the IS in AVC analytical method because the method of extraction (protein precipitation using ACN) could be utilized for AVC and GSB, which resulted in recoveries of 101.4 ± 2.5% and 103.3 ± 4.3%, respectively. The elution times of GSB and AVC were 1.25 and 1.85 min, respectively, indicating good resolution. Both AVC and GSB are anti-cancer drugs that are not prescribed simultaneously; so, the established LC-MS/MS analytical method could be utilized for pharmacokinetic studies or the therapeutic drug monitoring of AVC. Figure 3 shows the MRM chromatogram of the lower quality control (LQC) of AVC at 15.0 ng/mL. In the current method, the MRM analyzer mode has two mass transitions (402→341 and 402→70), which is more selective compared to SRM with one mass transition (402→341) [14].

### 2.3. Bioanalytical Method Validation

#### 2.3.1. Specificity

There was a good separation of AVC and GSB (IS) chromatographic peaks (Figure 4). There was no noticeable interference from the HLM matrix at AVC and GSB retention times (Figure 4), revealing the LC-MS/MS analytical method specificity. In the control samples (positive and negative) of MRM chromatograms, no carry-over effect of AVC or GSB was seen.

#### 2.3.2. Sensitivity and Linearity

The established LC-MS/MS analytical method showed a linearity in the range of 5 ng/mL to 500 ng/mL, which is enough for the current metabolic stability application. The linear regression equation for the constructed calibration curve is y = 164.1x − 123.4 and the correlation coefficient (R^2^) is 0.9984. The specifications of the constructed curve are 1/x, weighting and excluding the origin. Six calibration curves with eleven calibration levels were performed, the RSD values were within <4.08%, and the back-calculations data for all level confirmed the linearity of the established LC-MS/MS method (Table 1). The lower limit of quantification (LLOQ) was 5.0 ng/mL.

#### 2.3.3. Precision and Accuracy

The precision and accuracy results for AVC quality controls were in agreement with the FDA guidelines for bioanalytical method validation [29]. The interday and intraday accuracy and precision of the established LC-MS/MS analytical method were −1.4% to 6.7% and −0.8% to 6.4%, respectively (Table 2). The current LC-MS/MS method exhibited reasonable precision and accuracy (˂6.65%) compared to the reported analytical methods (˂13%) [14,16].

#### 2.3.4. Matrix Effects of HLMs and AVC Extraction Recovery

The extraction recovery of the AVC QCs was 100.9 ± 2.1% and (RSD < 2.1%) (Table 2). The GSB extraction recovery was 102.1 ± 3.7%. The absence of a matrix effect on AVC or GSB parent ionization was confirmed by injecting two sets of HLM matrices. Set 1 was spiked with AVC LQC (15 ng/mL) and GSB (100 ng/mL), whereas set 2 was prepared by replacing the HLM matrix with the mobile phase. The HLMs matrix containing AVC and GSB exhibited a matrix effect (ME) of 100.7± 2.0% and 102.5 ± 3.7%, respectively. The IS-normalized ME was 0.99 (within the allowed range). So, these outcomes confirmed that the HLMs matrix had no significant influence on either AVC or GSB parent ionization inside the ESI source. The mean extraction recovery of AVC in the established LC-MS/MS method was 101.4 ± 2.5%, which is better than the reported analytical method (88.6–90%) [14,15].

### 2.4. Metabolic Stability

AVC (1 µM/mL) was used in HLMs matrix incubation, as it should be less than the Michaelis–Menten constant and the concentration level would represent an acceptable value between the sensitivity of the analytical system and the saturation of the microsomal enzymes. HLMs protein concentration was used at 1 mg of microsomal protein in each mL of incubation mixture to avoid nonspecific protein binding [30,31]. The AVC concentration in the incubated samples was calculated using a linear regression equation generated from a freshly constructed calibration curve for AVC standards in HLMs matrix. The AVC metabolic stability curve was established by plotting the percentage of AVC remaining (*y*-axis) against the incubation time (*x*-axis) (Figure 5A). The linear part of the metabolic curve (0–30 min) was utilized to construct another curve of the Ln of the % AVC remaining against the incubation time (0–30 min) (Figure 5B), which generates a linear regression equation of y = −0.0269x + 4.559, with R^2^ = 0.9875 and the slope (0.0269) representing the rate constant of AVC metabolic reaction (Table 3). The slope was 0.0269; therefore, the in vitro t_1/2_ was 25.77 min AVC CL_int_ was calculated according to the in vitro t_1/2_ method; therefore, the Cl_int_ of AVC was 26.9 µL/min/mg [32]. From these results, it can be predicted that AVC is a medium extraction ratio drug able to ensure moderate accumulation in the body and possibly result in good bioavailability, compared with other tyrosine kinase inhibitors (e.g., dacomitinib). Using simulation software and Cloe PK, these results can also be utilized to predict the in vivo pharmacokinetics of AVC [33].

## 3. Chemicals, Instruments and Methods

### 3.1. Chemicals

Alvocidib (Synonyms: HMR-1275; 99.72%) and glasdegib (Synonyms: PF-04449913; 99.25%) were purchased from MedChem Express (Princeton, NJ, USA). Pooled HLMs (M0567) were purchased from the Sigma-Aldrich company (St. Louis, MO, USA) and kept at −70 °C until the time of use in order to maintain their stability and viability. All solvents that were used in the current LC-MS/MS analytical method were of HPLC chromatographic grade, which are suitable for the current work. All chemicals and reference powders were of an AR grade suitable for the analytical method development application. HPLC-grade water was obtained from the in-house Milli-Q filtration system, which was procured from the Millipore company (Billerica, MA, USA). Acetonitrile (ACN), ammonium formate (NH_4_COOH), formic acid (HCOOH) and dimethyl sulfoxide (DMSO) were purchased from the Sigma-Aldrich company (St. Louis, MO, USA).

### 3.2. In Silico Software for AVC Metabolic Lability Assessment

The in silico metabolic lability of alvocidib was assessed by utilizing the P450 metabolism module of the StarDrop software package (version 6.4), which was procured from Optibrium Ltd. (Cambridge, MA, USA). The results revealed the lability and regioselectivity of atoms towards P450 metabolism and were summarized as composite site lability (CSL) values, which reveal the degree of metabolic lability of the atoms [34,35,36]. The in silico experiment was performed via in vitro work or using analytical method development in order to evaluate the importance of that type of method on the target analytes, which ultimately resulted in saving time, money and effort for practical in vitro experiments.

To estimate AVC susceptibility to metabolism, the site labilities of each atom were estimated and collected to compute the “composite site lability” (CSL), revealing the overall metabolic lability (stability) of the AVC. This was calculated from the combined rates of metabolism for all atoms on the analyte:

CSL =ktotalktotal+kw
where *k_w_* is the rate constant for water formation.

CSL is a crucial factor that demonstrates the metabolic rate of AVC, and the SMILES format, i.e., (CN1CC[C@@H]([C@@H](C1)O)C2=C(C=C(C3=C2OC(=CC3=O)C4=CC=CC=C4Cl)O)O), of AVC was uploaded to the StarDrop software for CSL calculation.

### 3.3. Instrumentation and Conditions

The LC-MS/MS system consisted of Acquity UPLC (Waters company; model code UPA), which was used for the chromatographic separation of analytes peaks, and the Acquity Triple Quadrupole Detector (TQD) MS (model code TQD), which was utilized for the detection and quantification of chromatographic peaks from the UPLC system. The UPLC-TQD chromatographic system was managed by MassLynx V4.1 software (Version 4.1, SCN 805). The tuning and optimization of parent ions and daughter ions was carried out using an IntelliStart module. The acquisition, processing and reporting of generated data was performed utilizing “QuanLynx module”, the quantification software, which is part of the MassLynx Software package. A nitrogen generator from the Peak Scientific company (Scotland, UK) was used to supply desolvation gas to help the ESI source to remove the mobile phase solution in order to enable conversion to a gaseous state. A rotary pump (PA, USA; SV40B) was utilized to establish and maintain a vacuum inside the TQD MS analyzer, and Argon gas (99.999%) was obtained locally to be utilized as a collision gas inside the quadrupole number 2 (collision cell) of the TQD MS analyzer.

The analysis of AVC (C_21_H_20_ClNO_5_) and GSB (C_21_H_22_N_6_O) was performed in the positive MRM mode (ESI+). The optimization of instrument parameters was performed using IntelliStart^®^ software in infusion mode (fluidics) to obtain reasonable sensitivity, mass accuracy, the enhancement of analyte peaks intensity and selectivity. The flow rate of the cone gas (Nitrogen) was 100 L/h. Argon, at 0.14 mL/min, was used as the collision gas to initiate the fragmentation of isolated parent ions into daughter ions in the collision cell (Quadrupole 2). Nitrogen (650 L/h) was used as drying gas at 350 °C. MRM quadrupole detection mode was used for analytes quantification and to improve the sensitivity and selectivity of the LC-MS/MS analytical method. The cone voltages for enhancement od parent ionization (AVC and GSB) were set to 40 (V) and 26 (V), respectively. The RF lens, capillary, extractor and voltages were set at 0.1 (V), 4 (kV) and 3.0 (V), respectively. The ESI temperature was optimized at 350 °C. The MRM mass transitions for AVC (Rt: 1.9 min) were 402→341 (CE: 24 V and CV: 40 V) and 402→70 (CE: 30 V and CV: 40 V) (Figure 3A). The GSB peak (Rt: 1.2 min) was estimated using the MRM transitions (parent to selected two daughters) 375→257 (CE: 12 V and CV: 18 V) and 375→96 (CE: 22 V and CV: 18 V) (Figure 3B). The dwell time for AVC and GSB mass transitions was 0.025 s. The MRM detection mode for the TQD MS analyzer was utilized for the quantification of AVC and GSB to avoid any interference from the HLM matrix, which elevated the sensitivity of the developed UPLC-TQD analytical method.

Optimized chromatographic parameters were used for separating AVC and glasdegib (GSB; IS), including the composition of mobile phase and the type of the column stationary phase. Formic acid (HCOOH) solution (0.1% in H_2_O) with a measured pH at 3.2 as ammonium formate solution (10 mM in H_2_O) at various pH using few drops of formic acid (4.0, 4.2, 4.5) caused the long running time and the tailing of the chromatographic peaks. The mobile phase consisted of 70% aqueous solution (0.1% HCOOH in H_2_O) and 30% ACN (organic phase) at a flow rate of 0.3 mL/min. Elevating the % organic phase (ACN) resulted in poor separation and peaks overlapping, while decreasing the % organic phase (ACN) resulted in unnecessarily long retention times. The different natures of stationary phases were tested for the efficiency regarding analytes separation through normal phase columns, such as HILIC columns, on which neither AVC nor GSB were chromatographically separated, and the best outcomes were obtained using a ZORBAX C18 column (Eclipse plus, 50 mm length, particle size 1.8 μm and internal diameter 2.1 mm) at 22 ± 2 °C. The run time and injection volume were 2 min and 5 µL, respectively.

### 3.4. AVC Working Solutions

AVC and GSB (IS) were reported to have a good solubility in DMSO at 33.33 mg/mL (82.94 mM; ultrasonication) and at ≥83.33 mg/mL (222.55 mM), respectively. The stock solutions of AVC and GSB were performed in DMSO at a concentration of 1 mg/mL in the reported solubility range. The stepwise dilution of AVC (1 mg/mL) and GSB (1 mg/mL) was performed utilizing the composition of the mobile phase to avoid an effect on the chromatographic peak such as tailing. AVC (1 mg/mL) was diluted ten times in three sequential steps to create AVC working solution 1 (WK1: 100 µg/mL), AVC WK2 (10 µg/mL) and AVC WK3 (500 ng/mL). GSB (1 mg/mL) was diluted in three sequential steps to obtain GSB WK3 (1 µg/mL).

### 3.5. AVC Calibration Levels

DMSO (2%), with slight warming (heat deactivates HLMs at 50 °C for 5 min), was used for deactivating HLMs before preparing the calibration standards to discard the metabolic effect of HLMs on the AVC concentration during LC-MS/MS analytical method validation [37,38,39]. An HLMs matrix after deactivation was utilized at a concentration of 1 mg protein/mL by diluting 30 µL HLMs to 1 mL with phosphate buffer (pH 7.4) containing 1 mM NADPH. Four concentrations of AVC calibration standards were selected as quality controls (QCs) for the bioanalytical method validation procedures: 5, 15, 150, and 400 ng/mL for a lower limit of quantification (LLOQ; 5 ng/mL), lower QC (LQC; 15 ng/mL), medium QC (MQC; 150 ng/mL), and high QC (HQC; 400 ng/mL). AVC standards were prepared by diluting AVC WK2 and AVC WK3 with the HLM matrix to prepare nine AVC concentration levels, including quality control levels, namely 5 (LLOQ), 15 (LQC), 50, 100, 150 (MQC), 200, 300, 400 (HQC), and 500 ng/mL, thus keeping the HLMs matrix not less than 90% to decrease the effect of dilution. This was carried out to confirm the similarity between the established calibration standards and the in vitro metabolic experiment samples. A total of 100 µL of GSB WK3 (IS) was added to 1 mL of the calibration levels and QCs.

The standard procedure for analyte extraction in the metabolic stability experiments is the protein precipitation technique, which was utilized for the extraction of AVC and GSB from the HLMs matrix through the addition of 2 mL of ACN to AVC concentration levels, then centrifugation for all samples were performed at 12 min at 14,000 rpm and 4 °C to clarify supernatants, followed by the filtration of mL of the supernatant to confirm the extract’s purity through a 0.22 µm syringe filter into 1.5 mL HPLC vials, and 5 µL was used for TQD MS analysis. Control samples: HLM matrix with IS (positive control) and HLMs matrix (negative control) were prepared following the same procedure as mentioned above. Controls (negative and positive) were utilized to confirm the absence of interference from HLMs matrix at the retention times of AVC and GSB. An AVC linear calibration curve was constructed by plotting the AVC nominal concentration (*x*-axis) versus the peak area ratio of the AVC to GSB (*y*-axis). The linear regression equation (y = ax + b) and the coefficient of variation (R^2^) were used to validate the linearity of the established LC-MS/MS analytical method.

### 3.6. Bioanalytical Method Validation

Analytical validation parameters for the established UPLC-TQD analytical method were evaluated according to the bioanalytical method validation guidelines, as defined by the FDA general regulations [40]. The method was validated for linearity, specificity, precision, sensitivity, accuracy, matrix effect, and extraction recovery.

#### 3.6.1. Specificity

The specificity of the established LC-MS/MS analytical method was evaluated by injecting six HLM matrix (blank) samples after protein precipitation extraction procedure. These samples were tested for chromatographic eluted peaks at the retention times of AVC and GSB at the data of MRM chromatograms of HLMs matrix (spiked AVC and GSB) samples. The MRM mode was used to reduce the influence of carryover on TQD MS system.

#### 3.6.2. Sensitivity and Linearity

The linearity and sensitivity of the LC-MS/MS analytical method were assessed using twelve constructed calibration curves for AVC, which were prepared and injected on the same day using calibration levels and quality control samples. Each calibration curve was constructed by plotting the nominal AVC concentration on the *x*-axis against the peak area ratio of AVC to GSB (IS) on the y-axis. The least-squares statistical method was used to compute the regression equation (y = ax + b) of the linear model. The LLOD and LLOQ were computed as stated by the pharmacopoeia using the standard deviation (SD) of the intercept and the slope as follows: LLOD = 3.3 × SD of intercept/slope and LLOQ = 10 × SD of intercept/slope.

#### 3.6.3. Precision and Accuracy

The intraday precision and accuracy of the established LC-MS/MS analytical methodology were computed by analyzing HLM matrix samples spiked with four AVC QCs on the same day. Interday calculations were performed in the same manner on three following days using the four QCs. To express the accuracy and precision of the method, % error (% error = [(average measured conc. − expected conc.)/expected conc.] × 100) and % relative SD (% RSD = SD ∗ 100/Mean) were used.

#### 3.6.4. Matrix Effect and Extraction Recovery

Extraction recovery and the matrix effect were evaluated using the four QCs levels. The % recovery of AVC from HLMs was evaluated by dividing the peak area ratio of A, QCs after extraction from the HLMs matrix, and B, QCs in the mobile phase (A/B × 100). The matrix effect (ME) on AVC or GSB ionization was verified by injecting two sets of matrices after extraction and purification into LC-MS/MS system. The HLM matrix (Set 1) was spiked with 15 ng/mL AVC (LQC) and 100 ng/mL GSB (IS), while the sample matrix (Set 2) was prepared by exchanging the HLMs with the mobile phase. The ME for AVC and GSB was calculated utilizing the following equation:ME of AVB or GSB=mean peak area ratioSet 1Set 2×100

The IS-normalized ME was calculated by the following equation:IS normalized ME=ME of AVCME of GSB IS

### 3.7. AVC Metabolic Stability

The profile of AVC metabolic stability, involving CL_int_ and in vitro t_1/2_, was established through the determination of the % remaining AVC concentration after incubation with the HLMs matrix (including active HLMs) for 50 min, according to the following steps. First step (initiation): the pre-incubation of 1 µM AVC with active HLMs matrix (without NADPH) at 37 °C for 10 min to attain the optimum temperature for enzymatic metabolic reactions. Second step (incubation): 1 mM NADPH was added as a cofactor for metabolic reactions. To confirm the outcomes, the same previous two steps were repeated three times [41]. Third step (termination): the incubation metabolic reaction was halted at certain time points, namely at 0, 2.5, 7.5, 15, 20, 30, 40, and 50 min, through the addition of 2 mL of ice-cold ACN, which acts as a protein denaturation for enzymes and precipitating agent in extraction procedures. A total of 100 µL of GSB WK3 (IS: 1 µg/mL) was added to the incubation mixture (active HLMs, NADPH, AVC) immediately prior to the halting of the metabolic reaction in order to avoid the influence of metabolic enzymes on the IS concentration. The extraction of the analytes (AVC and GSB) was performed as previously mentioned. Data analysis and processing was performed using the QuanLynx, which included with MassLynx 4.1 Software. The AVC concentration at certain time points was calculated, and the AVC metabolic stability curve was constructed. The AVC concentration at 0 min was considered to be 100% and the remaining AVC % was plotted against time. From this curve, the time points that exhibited linearity were selected to construct another curve, which showed the natural logarithm of the % remaining AVC for the selected points against time, thus revealing the slope and the rate constant of AVC metabolism, which was used to calculate the in vitro t_1/2_ (min) with the following equation:In−vitro t1/2=ln2Slope

Next, CL_int_ (µL/min/mg) was calculated utilizing the following equation [34]:$${\mathrm{CL}}_{\mathrm{int},\;}=\frac{0.693}{In-vitro\;t_{1/2}}\;.\frac{µL\;\mathrm{incubation}}{\mathrm{mg}\;\mathrm{microsomes}}\;$$

CL_int_ was then converted to in vivo clearance, utilizing the average liver weight and HLM protein concentration per gram of liver, as cited in the literature [42].

## 4. Conclusions

A bioanalytical LC-MS/MS analytical method was developed for AVC quantification in the HLMs matrix with the application of a metabolic stability estimation. The LC-MS/MS analytical method demonstrated eco-friendliness (short run time and less organic solvent), good sensitivity and accuracy, fast analysis, and high recovery. Our results revealed that AVC displayed in vitro t_1/2_ values of 25.77 min and a moderate CL_int_ (26.9 µL/min/mg), suggesting a moderate rate of clearance from the body. Therefore, acceptable in vivo bioavailability is expected. Based on these results, we predict that AVC could be administered to patients without the effect of the dose gathering inside the human blood, instead resulting in fast excretion through the liver. The in vitro metabolic experiment data were utilized to verify the outcomes of the in silico expectations. The in silico results from the P450 metabolism model matched the results generated from the in vitro metabolic incubations; therefore, the in silico software can be used to predict the metabolic stability of the drugs, saving time and resources. Further studies are necessary to examine this approach for in vivo therapeutic drug monitoring.

## Figures and Tables

**Figure 1 molecules-28-02368-f001:**
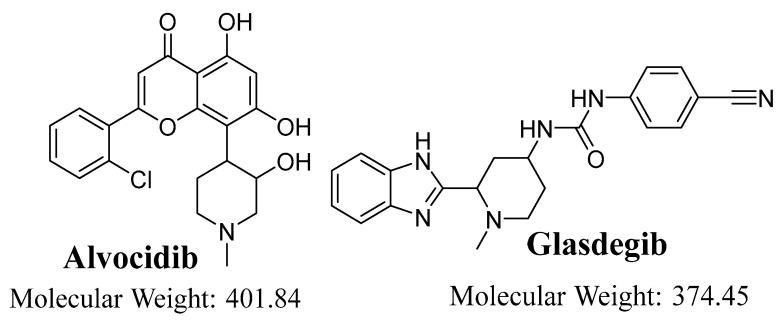
Chemical structures of alvocidib and glasdegib (IS).

**Figure 2 molecules-28-02368-f002:**
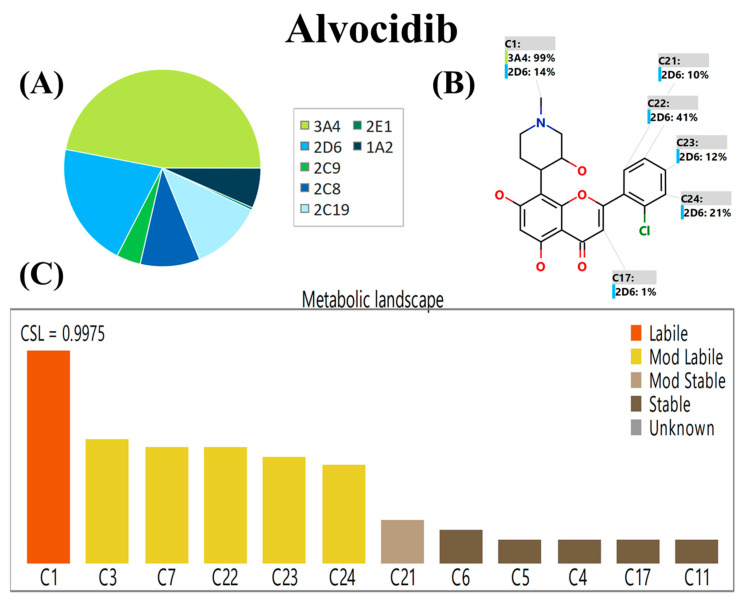
WhichP450 model proposed the major metabolizing isoform (CYP3A4) for AVC metabolism as approved by the pie chart (**A**). Regioselectivity map indicates the proposed sites of metabolism for AVC (**B**). Metabolic landscape showing CSL of AVC (0.9975) revealed the high metabolic rate (**C**).

**Figure 3 molecules-28-02368-f003:**
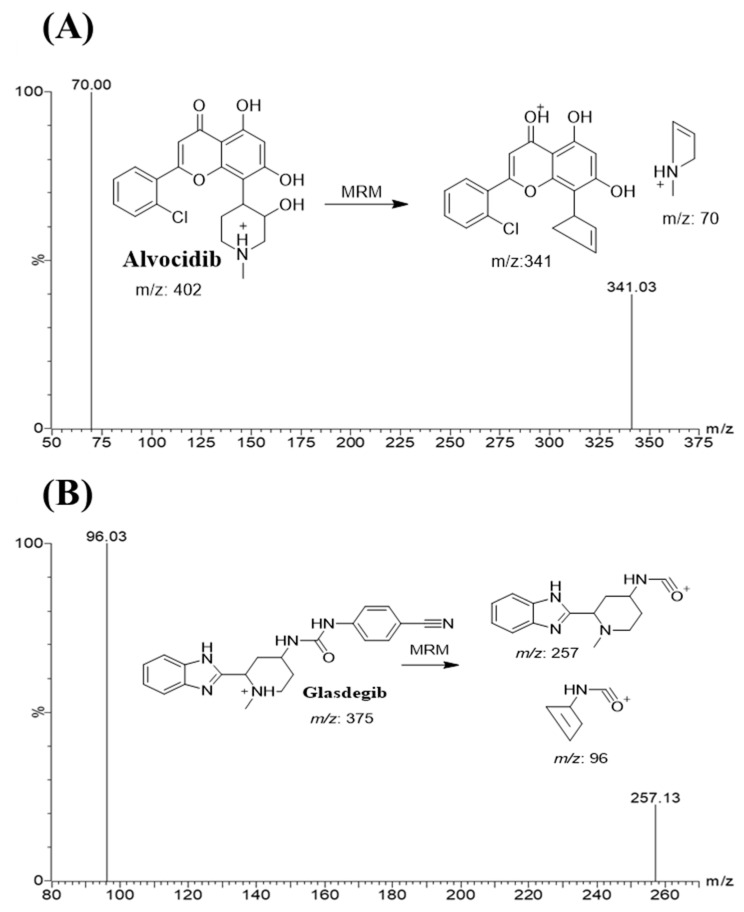
MRM mass spectra of alvocidib (AVC) (**A**) and glasdegib (GSB; internal standard; IS) (**B**) showing the predicted fragmentation pattern.

**Figure 4 molecules-28-02368-f004:**
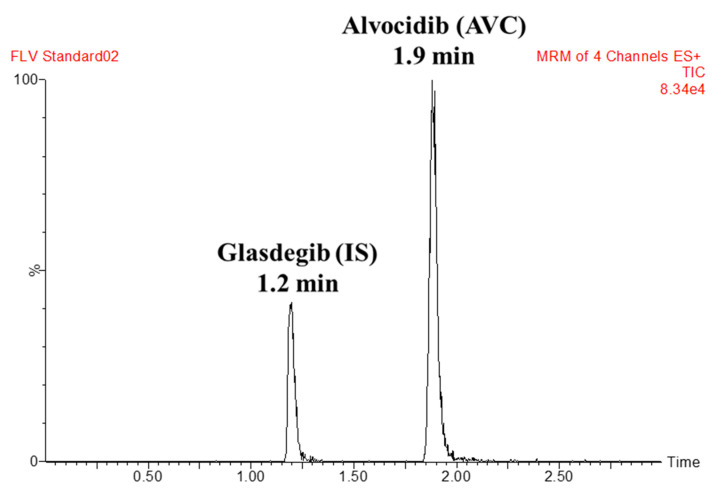
MRM chromatogram of the AVC LQC (15 ng/mL).

**Figure 5 molecules-28-02368-f005:**
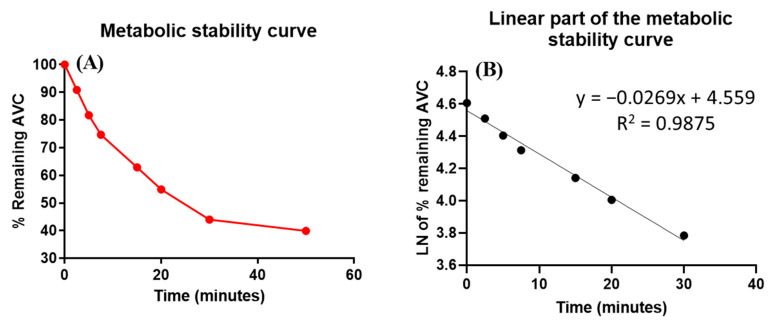
The metabolic stability curve of AVC in HLM (**A**) and the regression equation of the linear part of the curve (**B**).

**Table 1 molecules-28-02368-t001:** AVC back-calculation of six replicates of the calibration standards.

AVC Nominal Concentrations (ng/mL)	Mean ^a^	SD	RSD (%) ^b^	% Error
5.0 (LLOQ)	5.3	0.2	3.3	6.4
15.0 (LQC) ^c^	15.3	0.3	2.3	1.9
50.0	51.2	2.1	4.1	2.4
100.0	101.2	3.6	3.5	1.2
150.0 (MQC) ^c^	148.8	3.9	2.6	−0.8
200.0	195.7	2.9	1.5	−2.2
300.0	299.1	2.8	0.9	−0.3
400.0 (HQC) ^c^	401.8	6.0	1.5	0.5
500.0	506.9	2.8	0.6	1.4

^a^ Average of six calibration curves. ^b^ The RSD values for the six calibration curves were within <4.08%. ^c^ LQC, lower quality control; MQC, medium quality control; HQC, high quality control.

**Table 2 molecules-28-02368-t002:** Intraday and interday (precision and accuracy) of the developed LC-MS/MS method.

AVC in HLM Matrix (ng/mL)	Intraday Assay *				Interday Assay **			
	5 (LLOQ)	15 (LQC)	150 (MQC)	400 (HQC)	5 (LLOQ)	15 (LQC)	150 (MQC)	400 (HQC)
Mean	5.3	15.3	148.8	401.8	5.2	15.2	148.0	399.5
SD	0.2	0.3	3.9	6.0	0.3	0.6	3.4	6.0
Precision (% RSD)	3.3	2.3	2.6	1.5	6.7	4.0	2.3	1.5
% Error	6.4	1.9	−0.8	0.5	3.4	1.6	−1.4	−0.1
Recovery (%)	106.4	101.9	99.2	100.5	103.4	101.6	98.6	99.9

* Mean of twelve repeats on the same day. ** Mean of six repeats over three days.

**Table 3 molecules-28-02368-t003:** Parameters of AVC metabolic stability curve.

Time (min)	Mean ^a^ (ng/mL)	X ^b^	LN X	Analytical Parameters
0.0	486.9	100.0	4.6	Regression equation: y = −0.0269x + 4.559
2.5	442.5	90.9	4.5	
5.0	398.0	81.7	4.4	R^2^ = 0.9875
7.5	363.5	74.7	4.3	
15.0	306.0	62.9	4.1	Slope: −0.0269
20.0	267.1	54.9	4.0	
30.0	214.1	44.0	3.8	t_1/2_: 25.8 min and
50.0	194.1	39.9	3.7	Cl_int_: 26.9 µL/min/mg

^a^ Average of three repeats. ^b^ X: Average of the percentage AVC remaining from three repeats.

## Data Availability

All data are available within the manuscript.

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
