# Peer review of "Development and Validation of a Rapid LC-MS/MS Method for Quantifying Alvocidib: In Silico and In Vitro Metabolic Stability Estimation in Human Liver Microsomes"

_molecules, 2023, doi:10.3390/molecules28052368_

Round 1

Reviewer 1 Report

In the manuscript A validated LC-MS/MS method for quantifying alvocidib: Metabolic stability estimation in human liver microsomes, the authors developed an LC-MS/MS for the estimation of the metabolic stability of alvocidib (AVC) in human liver microsomes. The author also used in silico calculation to predict the metabolic stability of AVC, which agreed with the experimental results using the established methods. The study of the metabolism process of AVC is valuable for clinical treatment.

However, I think this work is NOT qualified to be published in MOLECULES for the follow reasons:

(1)   Glasdegib (GSB) is quite different with AVC in structure, and thereby is not suitable as an internal standard for the quantification of AVC.

(2)   No comparation of the current method with previous works is presented in the Results and Discussion section.

(3)   No significant innovation in terms of methodology is shown in the manuscript.

Other minor issues on the manuscript includes:

(1)   The organization of the structure of the Introduction section needs improvement. A brief introduction of the unsolved problems in previous works and a description of the current method are needed.

(2)   Abbreviations should be defined upon first use in the main text. Such as LC-MS/MS, HPLC-UV, ME, et al.

(3)   There are a few grammatical errors. Please check carefully.

(4)   Avoid unnecessary details in the main text, such as the systematic name of AVC (in Line 77-78).

(5)   Improper citation format in Line 68 and 72.

Author Response

Dear Supreeya Srisuk, Assistant Editor

Molecules Journal

Manuscript ID: molecules-2232294
Editor Comments”

Please revise the manuscript according to the referees' comments and upload the revised file within 10 days.

** Please use the version of your manuscript found at the above link (or attachment) for your revisions. **

(I) Please check that all references are relevant to the contents of the manuscript.

(II) Any revisions to the manuscript should be marked up using the “Track Changes” function if you are using MS Word/LaTeX, such that any changes can be easily viewed by the editors and reviewers.

(III) Please provide a cover letter to explain, point by point, the details of the revisions to the manuscript and your responses to the referees’ comments.

(IV) If you found it impossible to address certain comments in the review reports, please include an explanation in your appeal.

(V) The revised version will be sent to the editors and reviewers.

Authors’ response

We thank the editor for this opportunity to improve our manuscript and be considered again for publication in Molecules Journal. We give below detailed answers to each question raised by reviewer # 1. All replies to the comments were highlighted in yellow color in the revised manuscript.

Reviewer # 1

Comments and Suggestions for Authors

In the manuscript A validated LC-MS/MS method for quantifying alvocidib: Metabolic stability estimation in human liver microsomes, the authors developed an LC-MS/MS for the estimation of the metabolic stability of alvocidib (AVC) in human liver microsomes. The author also used in silico calculation to predict the metabolic stability of AVC, which agreed with the experimental results using the established methods. The study of the metabolism process of AVC is valuable for clinical treatment.

We appreciate the reviewer’s words and his/her suggestions to improve our manuscript. We give below our answer to his/her concerns.

Point # 1  

(1)   Glasdegib (GSB) is quite different with AVC in structure, and thereby is not suitable as an internal standard for the quantification of AVC.

Authors’ response

GSB was selected as the IS in AVC analytical method because of the following reasons:

1-The method of extraction (protein precipitation using ACN) could be utilized for AVC and GSB that resulted recoveries of 101.4 ± 2.54 % and 103.32 ± 4.32 %, respectively.

2- The elution times of GSB and AVC were 1.25 and 1.85 min, respectively, indicating good resolution.

3- Both AVC and GSB are anti-cancer drugs that are not prescribed simultaneously; so, the established LC-MS/MS analytical method could be utilized for pharmacokinetic studies or therapeutic drug monitoring of AVC.

4- There are many reported articles that used IS that is no similar in structure to the analyte being analyzed.

The following paragraph was updated in the manuscript:

GSB was selected as the IS in AVC analytical method because the method of extraction (protein precipitation using ACN) could be utilized for AVC and GSB that resulted recoveries of 101.4 ± 2.54 % and 103.32 ± 4.32 %, respectively. The elution times of GSB and AVC were 1.25 and 1.85 min, respectively, indicating good resolution. Both AVC and GSB are anti-cancer drugs that are not prescribed simultaneously; so, the established LC-MS/MS analytical method could be utilized for pharmacokinetic studies or therapeutic drug monitoring of AVC.

 Point # 2  

(2)   No comparation of the current method with previous works is presented in the Results and Discussion section.

Authors’ response

Comparing to the other methods, we added the following paragraph to the introduction:

The current LC-MS/MS method exhibited reasonable precision and accuracy (Ë‚6.65 %) compared to the reported analytical methods (Ë‚13 %) [14, 16]. The mean extraction recovery of AVC in the established LC-MS/MS method was 101.4 ± 2.54 % that is better than the reported analytical method (88.6–90 %) [14, 15]. In addition, the published method utilized a selective reaction monitoring with one mass transition (SRM: 402 → 341) for detection and a gradient mobile phase system for analysis [14] which is less selective than multiple reaction monitoring (MRM: 402→341 and 402→70) and isocratic elution mobile phase system that were utilized in the current LC-MS/MS method. No LC-MS/MS analytical method has been reported for metabolic stability quantification of AVC in human liver microsomes (HLMs).

Comparing to the other methods, we added many sentences highlighted in the results and discussion highlighted in yellow color.

Point # 3  

(3)   No significant innovation in terms of methodology is shown in the manuscript.

Authors’ response

We updated the manuscript with many new information about the current work including the use of in silico software to support the practical application.

Also we added and compared the current method with the previously reported methods:

The current LC-MS/MS method exhibited reasonable precision and accuracy (Ë‚6.65 %) compared to the reported analytical methods (Ë‚13 %) [14, 16]. The mean extraction recovery of AVC in the established LC-MS/MS method was 101.4 ± 2.54 % that is better than the reported analytical method (88.6–90 %) [14, 15]. In addition, the published method utilized a selective reaction monitoring with one mass transition (SRM: 402 → 341) for detection and a gradient mobile phase system for analysis [14] which is less selective than multiple reaction monitoring (MRM: 402→341 and 402→70) and isocratic elution mobile phase system that were utilized in the current LC-MS/MS method. No LC-MS/MS analytical method has been reported for metabolic stability quantification of AVC in human liver microsomes (HLMs).

Other minor issues on the manuscript includes:

Point # 1  

(1)   The organization of the structure of the Introduction section needs improvement. A brief introduction of the unsolved problems in previous works and a description of the current method are needed.

Authors’ response

We updated the following paragraphs giving more description to the current method in the introduction.

The current LC-MS/MS method exhibited reasonable precision and accuracy (Ë‚6.65 %) compared to the reported analytical methods (Ë‚13 %) [14, 16]. The mean extraction recovery of AVC in the established LC-MS/MS method was 101.4 ± 2.54 % that is better than the reported analytical method (88.6–90 %) [14, 15]. In addition, the published method utilized a selective reaction monitoring with one mass transition (SRM: 402 → 341) for detection and a gradient mobile phase system for analysis [14] which is less selective than multiple reaction monitoring (MRM: 402→341 and 402→70) and isocratic elution mobile phase system that were utilized in the current LC-MS/MS method. No LC-MS/MS analytical method has been reported for metabolic stability quantification of AVC in human liver microsomes (HLMs).

In the developed LC-MS/MS method, isocratic mobile phase was used. Run time and flow rate was 3 min. and 0.3 mL/min., respectively were used, so the methods was rapid and eco-friendly. Moreover, the calibration curve was linear in the range of 5 to 500 ng/mL. AVC was tested for its metabolism lability in the HLMs using StarDrop's WhichP450 model before the initiation of the practical experiment to confirm the importance of de-veloping an LC-MS/MS method and to save resources and time. The established LC-MS/MS method was applied for assessment of the intrinsic clearance (Clint) and in vitro half-life (t1/2) of AVC. In silico StarDrop's WhichP450 model software and in vitro LC-MS/MS experiments were used for the assessment of AVC metabolic stability so as to give more information about the metabolic rate of AVC and to allow in vivo bioavailability assessment.

Point # 2  

(2)   Abbreviations should be defined upon first use in the main text. Such as LC-MS/MS, HPLC-UV, ME, et al.

Authors’ response

We revised the whole manuscript. All abbreviation were defined upon first use.

Point # 3  

(3)   There are a few grammatical errors. Please check carefully.

Authors’ response

The whole manuscript was revised carefully for grammatical errors.

Point # 4  

(4)   Avoid unnecessary details in the main text, such as the systematic name of AVC (in Line 77-78).

Authors’ response

We removed many unnecessary details. We arranged the whole manuscript.

Point # 5  

(5)   Improper citation format in Line 68 and 72.

Authors’ response

Sorry for the typo mistake during manually editing the references. The typo mistake was corrected as requested. The correct citations were updated in the revised version of the manuscript as the following:

The current work was performed to develop an LC-MS/MS analytical method for AVC estimation in a HLMs matrix. Our group published other analytical LC-MS/MS chromatographic methods that were applied previously in the computation of the intrinsic clearance (CLint) and in-vitro half-life (t1/2) of other tyrosine kinase inhibitors [17]. Such study can reveal more information about the AVC metabolic destiny and allow the expectation of its in vivo bioavailability utilizing three models: dispersion models, venous equilibrium, and parallel tubes [18, 19]. Here, the AVC in-vitro CLint and t1/2 in HLM were calculated by an ‘in-vitro t1/2’ approach using the ‘well-stirred’ model [20, 21]. If the drug exhibited a high metabolic rate, it will have a low in vivo bioavailability and short window of action [22-25].

Kind regards

Dr. Mohamed Attwa

Dept. Pharm. Chem., College of Pharmacy, King Saud University,

P.O. Box 2457,

Riyadh, 11451, Saudi Arabia

mzeidan@ksu.edu.sa

Reviewer 2 Report

This article presents an LC-MS/MS method for alvocidib and an in silico study of metabolic stability.  The method presented is sound and has an emphasis on a rapid run time to be eco-friendly.  I suggest the word rapid be added to the title.  The manuscript is similar to  https://doi.org/10.3390/molecules28041641 which is by at least one of the same authors.  Overall, the citations should be checked by the authors to line up with refrences.  I think there are an adequate number of refernces.

Major concerns: Most citations are correctly formatted, but the citations in line 68 and line 72 are combination text numbers with superscript numbers.  I'm not sure that refernce 1 and 5 were intended to cited on line 68 and 72.  They don't seem to be related to the text in the sentence.

In 2.1, I think there should be more discussion about how the metabolic stability study is done.  Probably the name of the software should be mentioned here, and CSL should be defined and spelled out the first time it appears in text.  

Minor revisions:

Line 18 "6.45 variability" Is this RSD %?  

Line 33 "split"

Line 43 AML is spelled out, but the abbreviation AML was introduced in line 31.  Just use the abbreviation here.

Line 35: take out the word "following"

Line 48: Change out "unluckily" with "unfortunately"  This is my personal preference.

Line 55: remove the phrase "that..." after [16] as this idea is siad in line 52.

Line 66 change out estimation with quantification.

Line 77 is instead of was

Line 102  LQC should be spelled out here or just say the concentration in ng/mL

Line 117 "LC-MS/MS"

Line 120 write something like "of the linear construction curve are 1/x weighting and excluding the origin.  I'm not sure what Axis trans: none is but it probably doesn't need to go in the text.  

Line 124 use LLQC to be consistent.  LLQC should be spelled out the first time it appears.

Table 1  LQC and MQC and HQC should be spelled out in a footnote in the table as this is the first time they appear.

section 3.2 You don't need instrument serial numbers.  and section 3.2 should be combined with 3.4.  The LC-MS/MS methods should appear similar to https://doi.org/10.3390/molecules28041641

Author Response

Dear Supreeya Srisuk, Assistant Editor

Molecules Journal

Manuscript ID: molecules-2232294
Editor Comments”

Please revise the manuscript according to the referees' comments and upload the revised file within 10 days.

** Please use the version of your manuscript found at the above link (or attachment) for your revisions. **

(I) Please check that all references are relevant to the contents of the manuscript.

(II) Any revisions to the manuscript should be marked up using the “Track Changes” function if you are using MS Word/LaTeX, such that any changes can be easily viewed by the editors and reviewers.

(III) Please provide a cover letter to explain, point by point, the details of the revisions to the manuscript and your responses to the referees’ comments.

(IV) If you found it impossible to address certain comments in the review reports, please include an explanation in your appeal.

(V) The revised version will be sent to the editors and reviewers. If one of the referees has suggested that your manuscript should undergo extensive English revisions, please address this issue during revision. We propose that you use one of the editing services listed at https://www.mdpi.com/authors/english or have your manuscript checked by a

native English-speaking colleague.

Authors’ response

We thank the editor for this opportunity to improve our manuscript and be considered again for publication in Molecules Journal. We give below detailed answers to each question raised by reviewer # 2. All replies to the comments were highlighted in yellow color in the revised manuscript.

Reviewer # 2

Comments and Suggestions for Authors

This article presents an LC-MS/MS method for alvocidib and an in silico study of metabolic stability.  The method presented is sound and has an emphasis on a rapid run time to be eco-friendly.  I suggest the word rapid be added to the title.  The manuscript is similar to  https://doi.org/10.3390/molecules28041641 which is by at least one of the same authors.  Overall, the citations should be checked by the authors to line up with refrences.  I think there are an adequate number of refernces.

Authors’ response

We appreciate the reviewer’s words and his/her suggestions to improve our manuscript. We give below our answer to his/her concerns.

The citation was revised and corrected all over the manuscript.

The title of the manuscript was changed to

Development and validation of a rapid LC-MS/MS method for quantifying alvocidib: In silico and In vitro Metabolic stability estimation in human liver microsomes.

Point # 1  

Major concerns: Most citations are correctly formatted, but the citations in line 68 and line 72 are combination text numbers with superscript numbers.  I'm not sure that refernce 1 and 5 were intended to cited on line 68 and 72.  They don't seem to be related to the text in the sentence.

Authors’ response

Sorry for the typo mistake during manually editing the references. The typo mistake was corrected as requested.

The correct citations were updated in the revised version of the manuscript as the following:

The current work was performed to develop an LC-MS/MS analytical method for AVC estimation in a HLMs matrix. Our group published other analytical LC-MS/MS chromatographic methods that were applied previously in the computation of the intrinsic clearance (CLint) and in-vitro half-life (t1/2) of other tyrosine kinase inhibitors [17]. Such study can reveal more information about the AVC metabolic destiny and allow the expectation of its in vivo bioavailability utilizing three models: dispersion models, venous equilibrium, and parallel tubes [18, 19]. Here, the AVC in-vitro CLint and t1/2 in HLM were calculated by an ‘in-vitro t1/2’ approach using the ‘well-stirred’ model [20, 21]. If the drug exhibited a high metabolic rate, it will have a low in vivo bioavailability and short window of action [22-25].

Point # 2  

In 2.1, I think there should be more discussion about how the metabolic stability study is done.  Probably the name of the software should be mentioned here, and CSL should be defined and spelled out the first time it appears in text. 

Authors’ response

We added the following sentences and figure to the revised version of manuscript:

P450 metabolism model of StarDrop’s software (Optibrium Ltd.; Cambridge, MA, USA) predicts the regioselectivity of metabolism by seven of the major drug metabolizing isoforms of Cytochrome P450 enzymes (CYP3A4, CYP2D5, CYP2C9, CYP1A2, CYP2C19, CYP2C8 and CYP2E1). The results are expressed as composite site lability (CSL) value revealing the AVC metabolic lability. To compute the metabolic lability of AVC, the individual atoms labilities can be collected to calculate the CSL revealing the overall metabolic stability for AVC following the equation 1:

        Equation (1)

as kw is the rate constant for water formation.

WhichP450 model predicts the major metabolizing isoform (CYP3A4) for AVC metabolism as approved by the pie chart (Fig. 2A). Regioselectivity map indicates predicted sites of metabolism for AVC (Fig. 2B). The metabolic landscape (Fig. 3C) predicts the AVC metabolic lability of the active sites to improve the understanding of the AVC metabolic rate.

Figure 2. WhichP450 model proposed the major metabolizing isoform (CYP3A4) for AVC metabolism as approved by the pie chart (A). Regioselectivity map indicates proposed sites of metabolism for AVC (B). Metabolic landscape showing CSL of AVC (0.9975) revealed the high metabolic rate (C).

Minor revisions:

Point # 1  

Line 18 "6.45 variability" Is this RSD %? 

Authors’ response

The sentence was corrected as the following:

The interday and intraday accuracy and precision of the established LC-MS/MS analytical method were -1.36 % to 6.65 % and -0.83 % to 6.42 %, respectively, confirming the reproducibility of the LC-MS/MS analytical method.

Point # 2  

Line 33 "split"

Authors’ response

The typo mistake is corrected as requested.

Point # 3  

Line 43 AML is spelled out, but the abbreviation AML was introduced in line 31.  Just use the abbreviation here.

Authors’ response

The typo mistake is corrected as requested.

Point # 4  

Line 35: take out the word "following"

Authors’ response

The word following was removed as requested.

Point # 5  

Line 48: Change out "unluckily" with "unfortunately"  This is my personal preference.

Authors’ response

The word "unluckily" was changed to "unfortunately"  as requested. It is more better for us also.

Point # 6  

Line 55: remove the phrase "that..." after [16] as this idea is siad in line 52.

Authors’ response

The sentence was corrected as requested:

The mean extraction recovery of AVC in the established LC-MS/MS method was 101.4 ± 2.54 % that is more better than the reported analytical method (88.6–90 %)

Point # 7  

Line 66 change out estimation with quantification.

Authors’ response

It was done as requested

Point # 8  

Line 77 is instead of was

Authors’ response

It was done as requested

Point # 9  

Line 102  LQC should be spelled out here or just say the concentration in ng/mL

Authors’ response

It was done as requested

Fig. 3 shows the MRM chromatogram of the lower quality control (LQC) of AVC at 15 ng/mL.

Point # 10  

Line 117 "LC-MS/MS"

Authors’ response

The typo mistake was corrected.

Point # 11  

Line 120 write something like "of the linear construction curve are 1/x weighting and excluding the origin.  I'm not sure what Axis trans: none is but it probably doesn't need to go in the text. 

Authors’ response

It was rewritten as requested

The specifications of the constructed curve are 1/x weighting and excluding the origin.

Point # 12  

Line 124 use LLQC to be consistent.  LLQC should be spelled out the first time it appears.

Authors’ response

The correct abbreviation is LLOQ, so changed LLQC to LLOQ all over the manuscript.

The lower limit of quantification (LLOQ) was 5 ng/mL.

Point # 13  

Table 1. LQC and MQC and HQC should be spelled out in a footnote in the table as this is the first time they appear.

Authors’ response

LQC and MQC and HQC was spelled out in a footnote in the table as requested:  

c LQC, lower quality control; MQC, medium quality control; HQC, high quality control.

Point # 14  

section 3.2 You don't need instrument serial numbers.  and section 3.2 should be combined with 3.4.  The LC-MS/MS methods should appear similar to https://doi.org/10.3390/molecules28041641

Authors’ response

We rearranged the manuscript as requested and we removed the serial number.

The LC-MS/MS system consisted of Acquity UPLC [Waters company; model code UPA], that was used for the chromatographic separation of analytes peaks, and Acquity Triple Quadrupole Detector (TQD) MS [model code TQD] that was utilized for detection and quantification of chromatographic peaks from the UPLC system. The UPLC-TQD chromatographic system was managed by MassLynx V4.1 software (Version 4.1, SCN 805). Tuning and optimization of parents’ ions and daughters’ ions were done using an IntelliStart module. Acquisition, processing and reporting of generated data were done utilizing ‘QuanLynx module', the quantification software, that is part of the MassLynx Software package. A nitrogen generator from Peak Scientific company (Scotland, UK) was used to supply desolvation gas to help ESI source in getting rid of mobile phase solution for converting to gaseous state. A rotary pump (PA, USA; SV40B) was utilized to establish and maintain vacuum inside the TQD MS analyser, Argon gas (99.999 %) was obtained locally to be utilized as a collision gas inside the quadrupole number 2 (collision cell) of the TQD MS analyser.

Analysis of AVC (C21H20ClNO5) and GSB (C21H22N6O) was performed in the positive MRM mode (ESI+). Optimizing of instrument parameters was tuned using IntelliStart® software in infusion mode (fluidics) to obtain reasonable sensitivity, mass accuracy, enhancement of analyte peaks intensity and selectivity. The flow rate of the cone gas (Nitrogen) was 100 L/h. Argon at 0.14 mL/min was used as the collision gas to initiate the fragmentation of isolated parent ions into daughter ions in the collision cell (Quadrupole 2). Nitrogen (650 L/h) was used as drying gas at 350°C. MRM quadrupole detection mode was used for analytes quantification and to improve the sensitivity and selectivity of the LC-MS/MS analytical method. The cone voltages for enhancement od parent ionization (AVC and GSB) were set to 40 (V) and 26 (V), respectively. The RF lens, capillary, extractor and voltages were set at 0.1 (V), 4 (kV), and 3.0 (V), respectively. The ESI temperature was optimized at 350 °C. The MRM mass transitions for AVC (Rt:1.9 min) were 402→341 (CE:24V and CV:40 V) and 402→70 (CE:30 V and CV:40 V) (Fig. 3A). The GSB peak (Rt:1.2 min) was estimated using the MRM transitions (parent to selected two daughters) 375→257 (CE:12 V and CV:18 V) and 375→96 (CE:22 V and CV:18 V) (Fig. 3B). The dwell time for AVC and GSB mass transitions was 0.025 s. The MRM detection mode for TQD MS analyser was utilised for the quantification of AVC and GSB to avoid anay interference from HLM matrix, which elevated the sensitivity of the developed UPLC-TQD analytical method.

Optimized chromatographic parameters were used for separating AVC and glasdegib (GSB; IS), including the composition of mobile phase and the type of the column stationary phase. Formic acid (HCOOH) solution (0.1 % in H2O) with a measured pH at 3.2 as ammonium formate solution (10 mM in H2O) at various pH using few drops of formic acid (4.0, 4.2, 4.5) caused the long running time and the tailing of the chromatographic peaks. The mobile phase consisted of 70 % aqueous solution (0.1 % HCOOH in H2O) and 30 % ACN (organic phase) at a flow rate of 0.3 mL/min. Elevating the % organic phase (ACN) resulted in the poor separation and peaks overlapping, while decreasing % organic phase (ACN) resulted in unnecessary longer retention times. Different nature of stationary phases was tested for the efficiency towards analytes separation, normal phase columns such as HILIC columns, on which neither AVC nor GSB were chromatographically separated, and best outcomes were obtained using a ZORBAX C18 column (Eclipse plus, 50 mm length, particle size 1.8 μm and internal diameter 2.1 mm) at 22±2 °C. The run time and injection volume were 2 min and 5 µl, respectively.

Kind regards

Dr. Mohamed Attwa

Dept. Pharm. Chem., College of Pharmacy, King Saud University,

P.O. Box 2457,

Riyadh, 11451, Saudi Arabia

mzeidan@ksu.edu.sa

Reviewer 3 Report

Major comments

The manuscript suffers for a lack of novelty. Authors only identify the matrix as the main difference from other methods. Even though the manuscript may be of interest for readers of Molecules, I encourage the authors to pursue more ambitious goals in the future.

Minor comments

Numerical results such as accuracy, precision, etc. should be rounded to the appropriate significant figures.

Author Response

Dear Michalina Lassak, Section Managing Editor

Molecules Journal

Manuscript ID: molecules- 2248281

Editor Comments”

Please revise the manuscript according to the referees' comments and upload the revised file within 10 days.

** Please use the version of your manuscript found at the above link (or attachment) for your revisions. **

(I) Please check that all references are relevant to the contents of the manuscript.

(II) Any revisions to the manuscript should be marked up using the “Track Changes” function if you are using MS Word/LaTeX, such that any changes can be easily viewed by the editors and reviewers.

(III) Please provide a cover letter to explain, point by point, the details of the revisions to the manuscript and your responses to the referees’ comments.

(IV) If you found it impossible to address certain comments in the review reports, please include an explanation in your appeal.

(V) The revised version will be sent to the editors and reviewers. If one of the referees has suggested that your manuscript should undergo extensive English revisions, please address this issue during revision. We propose that you use one of the editing services listed at https://www.mdpi.com/authors/english or have your manuscript checked by a

native English-speaking colleague.

Authors’ response

We thank the editor for this opportunity to improve our manuscript and be considered again for publication in Molecules Journal. We give below detailed answers to each question raised by reviewer # 3. All replies to the comments were highlighted in yellow color in the revised manuscript.

Reviewer # 3

Comments and Suggestions for Authors

Authors’ response

We appreciate the reviewer’s words and his/her suggestions to improve our manuscript. We give below our answer to his/her concerns

Major comments

The manuscript suffers for a lack of novelty. Authors only identify the matrix as the main difference from other methods. Even though the manuscript may be of interest for readers of Molecules, I encourage the authors to pursue more ambitious goals in the future.

Authors’ response

Thank you for your advice, we are trying to add more applications and goals to our work in the future. We redesigned the whole manuscript trying to enhance the current work. We updated the manuscript with many new information about the current work including the use of in silico software to support the practical application. Also we added and compared the current method with the previously reported methods

Minor comments

Numerical results such as accuracy, precision, etc. should be rounded to the appropriate significant figures.

Authors’ response

We adjusted the significant figures all over the manuscript.

Kind regards

Dr. Mohamed Attwa

Dept. Pharm. Chem., College of Pharmacy, King Saud University,

P.O. Box 2457,

Riyadh, 11451, Saudi Arabia

mzeidan@ksu.edu.sa

.

Round 2

Reviewer 1 Report

The manuscript has addressed the reviewer comments and can now be accepted.